

# The relationship between MAPK signaling pathways and osteogenic differentiation of periodontal ligament stem cells: a literature review

Xuanning Liu[1,*], Wanqing Zhao[1,*], Yanhui Peng[1], Na Liu[2] and Qing Liu[1]

[1] Hebei Key Laboratory of Stomatology, Hebei Technology Innovation Center of Oral Health, School and Hospital of Stomatology, Hebei Medical University, Shijiazhuang, Hebei Province, China
[2] Department of Preventive Dentistry, School and Hospital of Stomatology, Hebei Medical University, Shijiazhuang, Hebei Province, China
[*] These authors contributed equally to this work.

## ABSTRACT

Periodontitis is a common oral disease that can lead to gingival inflammation, development of periodontal pockets, resorption of the alveolar bone, and the loosening and eventual loss of teeth. The optimal outcome of periodontitis treatment is maximum regeneration and functional reconstruction of periodontal tissues after control of infection and elimination of inflammation. Since both the self-healing ability of alveolar bone and the efficacy of traditional treatment methods are very limited, stem cell-based tissue regeneration engineering has received more and more attention from scholars. The best cells for periodontal tissue regeneration have been well examined, and these are called periodontal ligament stem cells (PDLSCs). The MAPK signaling pathways, including the ERK1/2, p38 MAPK, JNK, and ERK5 signaling pathways, are very complex and highly conserved tertiary kinase signaling pathways. These pathways are closely related to the osteogenic differentiation of PDLSCs, and this paper provides an overview of the research on the MAPK signaling pathways and the osteogenic differentiation of PDLSCs.

## INTRODUCTION

Periodontitis is a chronic inflammatory disease that affects the periodontal tissues (gingiva, periodontal membrane, cementum, and alveolar bone), leading to gingival inflammation, development of periodontal pockets, resorption of the alveolar bone, and, severe cases, the loosening and eventual loss of teeth (*Kwon, Lamster & Levin, 2021*). Controlling infection, reducing inflammation, and achieving the regeneration and functional rebuilding of periodontal tissue are the main objectives of treating periodontitis. Conventional treatments (supragingival scaling, subgingival scaling, and root planing) can combat pathogens to a certain extent but have limitations in restoring periodontal tissue morphology and function. In recent years, stem cell-based tissue engineering has emerged as a promising approach to treating periodontitis and achieving periodontal regeneration.

Corresponding authors
Na Liu, liuna@hebmu.edu.cn
Qing Liu, liuqing@hebmu.edu.cn

Periodontal ligament stem cells (PDLSCs) are mesenchymal stem cells with high proliferative capacity, low immunogenicity, and multidirectional differentiation potential. In 2004, *Seo et al. (2004)* successfully extracted human periodontal ligament stem cells (hPDLSCs) from adult, healthy periodontal tissues for the first time. Under appropriate culture conditions, PDLSCs can develop into fibroblasts, osteoblasts, chondrocytes, and adipocytes. PDLSCs not only maintain periodontal tissue homeostasis but also contribute to periodontal tissue regeneration, offering advantages in bone remodeling compared to other stem cells, and are thought to be the most promising seed cells in the periodontitis therapy and periodontal tissue regeneration engineering (*Morsczeck & Reichert, 2017*).

Mitogen-activated protein kinases (MAPK) are a broad family of serine-threonine kinases. They are crucial messengers that carry messages from the cell's outside to the nucleus. The MAPK signaling pathways are key regulators of cell proliferation, apoptosis, and differentiation. They are divided into four subfamilies: the extracellular signal-regulated kinase1/2 (ERK1/2) signaling pathway, the p38 mitogen-activated protein kinases (p38 MAPK) signaling pathway, the c-Jun N-terminal kinases (JNK) signaling pathway, and the extracellular signal-regulated kinase 5 (ERK5) signaling pathway (*Zhan, Li & Zhou, 2021*).

It is well known that there is a close relationship between the MAPK signaling pathways and bone formation (*Artigas et al., 2014*; *Greenblatt et al., 2022*; *Jiang & Tang, 2018*). At present, a large number of studies have proved that MAPK signaling pathways can affect the osteogenic differentiation of PDLSCs, but most of the studies have only verified that MAPK signaling pathways can affect the osteogenic differentiation of PDLSCs *in vitro* mineralization experiments using certain factors. There is no article that systematically classifies and summarizes these factors and discusses how the MAPK signaling pathways affect the osteogenic differentiation of PDLSCs. Therefore, this paper reviews the relationship between the MAPK signaling pathways and osteogenic differentiation of PDLSCs, summarizes the currently known relevant factors that affect the osteogenic differentiation of PDLSCs by activating the MAPK signaling pathways, and provides research ideas for the better application of PDLSCs in periodontal tissue regeneration. Clinicians and researchers in the field of dentistry and in the field of tissue engineering are provided with references and support.

## SURVEY METHODOLOGY

The following four databases were thoroughly searched to locate all relevant studies: MEDLINE (*via* PubMed), Embase, Cochrane Library, and Web of Science. Additionally, manual searches were completed by reviewing the reference lists of relevant articles. Articles, reviews, editorials, research snapshots, and systematic evaluations covering all topics included in this review, such as the MAPK signaling pathway and osteogenic differentiation of periodontal ligament stem cells, met the inclusion criteria for this review. "MAPK signaling pathway" and "osteogenic differentiation of periodontal ligament stem cells" were the keywords used in the search method. The search strategy was as follows: "("MAPK signaling pathway" (Mesh)) and ((Osteogenic differentiation of periodontal ligament stem cells) or (Osteogenic differentiation))" and "(("MAPK signaling pathway"

(Mesh)) and ("Osteogenic differentiation" (Mesh)),'' which was developed for MEDLINE and modified for the other databases. A systematic literature search was performed by combining various keywords such as "herbal medicine", "hormones", and "physical factors.''. To ensure the relevance and quality of the articles, several filters were applied during the search process. Articles published before 2010 were excluded to focus on the latest advances in the field.

Priority was given to articles published in high-impact journals within the fields of cell biology, regenerative medicine, and dentistry. Journals such as "Stem Cells", "Journal of Bone and Mineral Research", and "Tissue Engineering" were specifically included in the search criteria, while lesser-known or predatory journals were excluded to maintain the integrity of the research.

In summary, a systematic approach was used to search the literature on the MAPK signaling pathway and osteogenic differentiation of periodontal ligament stem cells, using targeted keywords and filtering based on publication year and journal quality strategies. This approach ensures the collection of a robust set of relevant and high-quality research articles for further analysis.

## LITERATURE REVIEW

### Biological effects of the MAPK signaling pathways

The MAPK signaling pathways comprise MAPK kinase kinases (MAP3K, MKKK), MAPK kinases (MAP2K, MKK), and MAPKs that are part of a highly conserved tertiary kinase cascade (*Ru, Pan & Zheng, 2023*) (Fig. 1). When cells are stimulated, these three kinases can be activated sequentially, and jointly regulate a variety of important cellular physiological or pathological processes such as cell growth, differentiation, adaptation to environmental stress, and inflammatory response (Fig. 2). MAPK pathways include ERK1/2 signaling pathway, p38 MAPK signaling pathway, JNK signaling pathway, and ERK5 signaling pathway. Different stimuli can activate one or more pathways, which in turn affect physiological functions of cells, including osteogenesis.

#### *ERK1/2 pathway signaling*

The ERK signaling pathways are classical MAPK signaling pathways, divided into Ras-dependent and Ras-independent pathways, with most pathways operating through the Ras-dependent mechanisms (*Lechuga et al., 2021*). Therefore, the ERK signaling pathways are also referred to as the Ras-Raf-MEK-ERK pathways. The upstream molecules of the ERK signaling pathways consist mainly of transmembrane receptor tyrosine kinases (RTKs), integrins, and G protein-coupled receptors (*Zhou et al., 2019*). Activation of transmembrane RTKs by extracellular stimulatory factors prompts growth factor receptor-bound protein 2 (GRB2) to attract Sos protein, forming the GRB2-Sos complex. The GRB2-Sos complex facilitates the binding of Ras protein to guanosine triphosphate (GTP), which activates Raf protein kinase, a MAP3K of the pathway. Raf then phosphorylates MAPK/ERK kinases (MEK), which in turn activate ERK. Ultimately, ERK activates downstream molecules located in the nucleus and cytoplasm (*Han & Yi, 2016*). There are two main isoforms of ERK proteins in this pathway: ERK1 and ERK2, which serve as

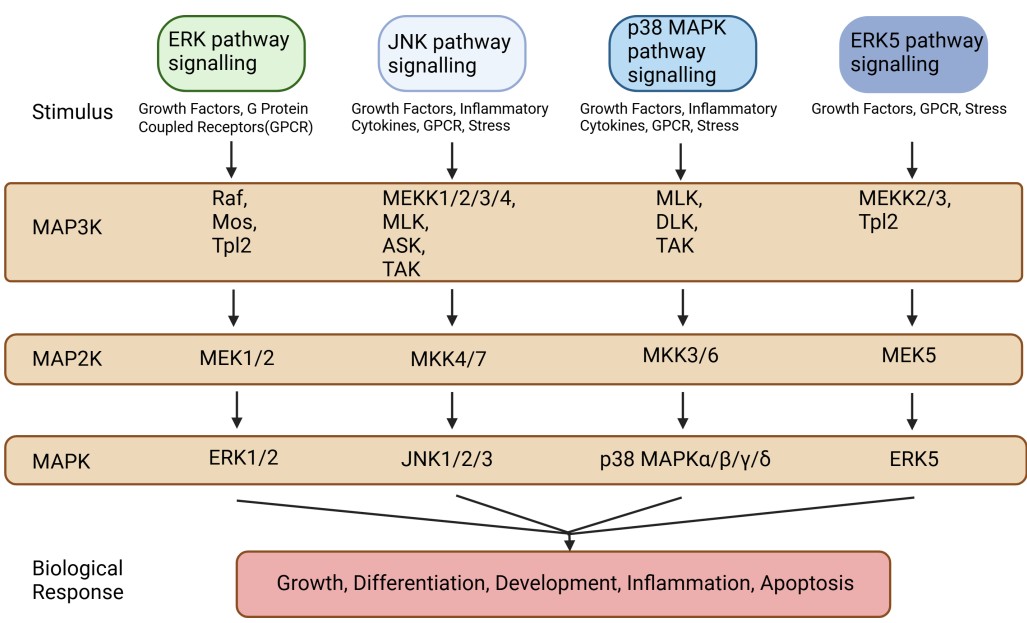

**Figure 1** **Types of MAP3Ks, MAP2Ks, and MAPKs.** MAPK pathways include the ERK1/2 signaling pathway, the p38 MAPK signaling pathway, the JNK signaling pathway, and the ERK5 signaling pathway. These MAPK signaling pathways include MAPK kinase kinases (MAP3K, MKKK), MAPK kinases (MAP2K, MKK), and MAPK, each of which contains multiple factors. Created in BioRender.

common signaling components in various signaling pathways. The biological effects of the ERK signaling pathways mainly include influencing cell proliferation and differentiation, regulating apoptosis, participating in cell migration, and regulating cellular sugar and lipid metabolism (*Liu & Luo, 2024*).

### p38 MAPK pathway signaling

p38 MAPK proteins are divided into six isoforms: p38α1, p38α2, p38β1, p38β2, p38γ, and p38δ. p38α1/α2 and p38β1/β2 are ubiquitously expressed in cells, p38γ is primarily found in skeletal muscle, and p38δ plays roles in the pancreas, kidney, testis, and small intestine (*Cuenda & Nebreda, 2009*). The p38 MAPK signaling pathway can be activated when cells are stimulated by physical factors (hypoxia, ultraviolet light, heat shock, ischemia-reperfusion), inflammatory factors, lipopolysaccharides, and other stimuli, following the MAPK tertiary enzymatic cascade pathway (*Li et al., 2016*). Activated p38 MAPK enters the nucleus and regulates the expression of relevant target genes. The biological effects of the p38 MAPK signaling pathway mainly include inducing cellular stress response, influencing cell proliferation and differentiation, regulating apoptosis, participating in cell migration, and participating in cellular immune response (*Li et al., 2024*).

### JNK pathway signaling

As members of the MAPK superfamily, JNK proteins are also referred to as stress-activated protein kinases (SAPKs) due to the fact that a range of extracellular stress signals have the ability to activate them. Three genes—JNK1, JNK2, and JNK3—encode the JNK

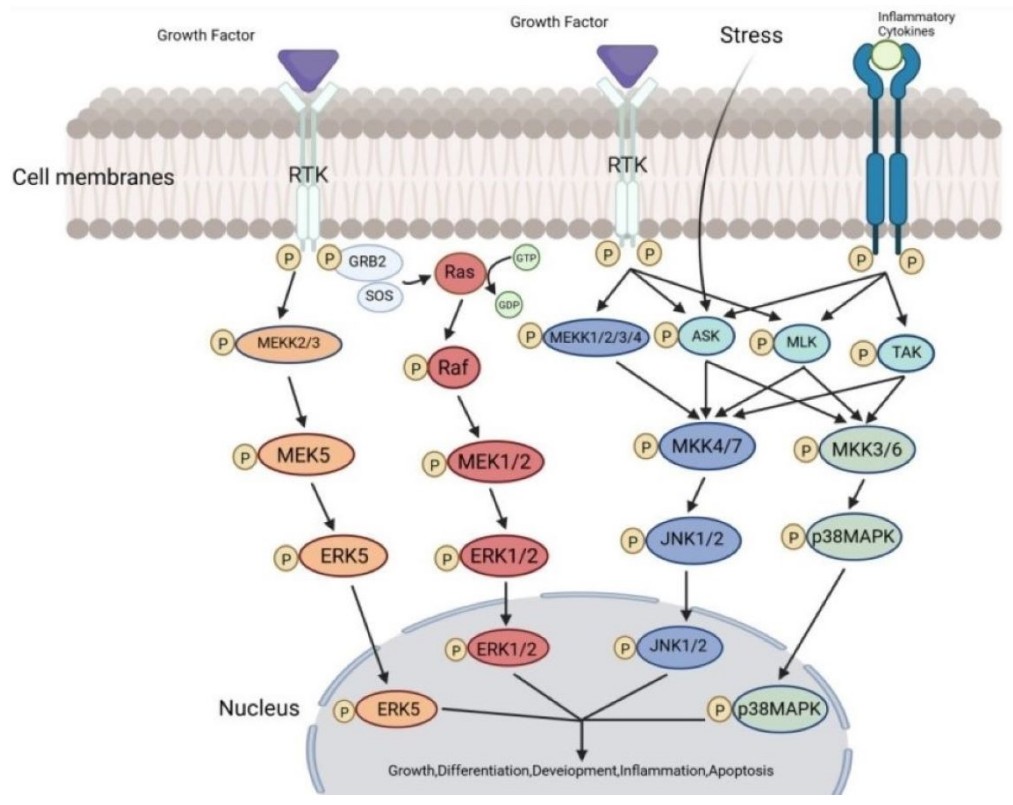

**Figure 2  The transmission process of MAPK signaling pathways in cells.** The MAPK pathway is a highly conserved tertiary kinase pattern. When the cell is stimulated, this MAP3K, MAP2K, and MAPK can be activated sequentially and jointly regulate a variety of important cellular physiological or pathological processes. Created in BioRender.

proteins. While JNK3 is mostly expressed in the brain, heart, and testes, JNK1 and JNK2 are expressed widely throughout the body (*Zhu et al., 2022*). The JNK signaling pathway's MAP3Ks mostly consist of TGF-β- activated protein kinase (TAK), mixed lineage kinase (MLK), apoptosis signal-regulating kinase (ASK), and MAPK/ERK kinase kinase 1/2/3/4 (MEKK 1/2/3/4). The MAP2Ks of the JNK signaling pathway are MAPK kinases 4/7 (MKK4/7). The JNK signaling pathway can be summarized as MEKK-1-4/M LK/ASK/TAK → MKK4/7 → JNK. Activated JNK can bisphosphorylate serine residues at positions 63 and 73 of the c-Jun protein, increasing the transcriptional activity of c-Jun and thereby inducing the expression of specific genes (*Weston & Davis, 2007*). The biological effects of the JNK signaling pathway mainly include regulating the cell cycle, affecting cell differentiation, regulating apoptosis, regulating immune cell development and function, and regulating nervous system function (*Castro-Torres et al., 2024*; *Li et al., 2024*).

### ERK5 pathway signaling

ERK5 consists of 816 amino acids and belongs to the same family as ERK1/2, sharing a similar threonine and tyrosine TEY phospho-module sequence with ERK1/2. Unlike ERK1/2, ERK5 has a large carboxyl terminus that contains a nuclear localization signal, two

proline-rich regions, and a transcriptional activation domain, making ERK5 nearly twice as long as other MAPK family members (*Hoang et al., 2017*). The ERK5 signaling pathway is an atypical MAPK pathway, but it also follows the same MAPK tertiary enzymatic cascade. Various stimuli activate MAPK/ERK kinase kinases 2/3 (MEKK2/3), then MEKK2/3 activate MAPK/ERK kinase 5 (MEK5), and finally, MEK5 activates ERK5. Notably, MEK5 is the only upstream kinase of ERK5, which remains highly specific for ERK5 and does not activate other members of the MAPK family even when overexpressed (*Pearson et al., 2001*). The biological effects of the ERK5 signaling pathway mainly include affecting cell proliferation and survival, regulating cell differentiation and angiogenesis, influencing cell migration, inducing cellular stress response, and regulating cellular sugar and lipid metabolism (*Wen et al., 2024*).

## MAPK signaling pathways are involved in osteogenic differentiation of PDLSCs

PDLSCs possess strong differentiation potential and an outstanding ability to repair bone defects among mesenchymal stem cells, making them the most promising candidates for periodontal tissue regeneration. Research has demonstrated that the osteogenesis-related transcription factors, such as runt-related transcription factor 2 (Runx2) and the osteoblast-specific transcription factor osterix (OSX), can be activated by the ERK1/2 signaling pathway to cause MSCs to differentiate into osteogenic tissues. It is one of the MAPK family pathways most closely associated with osteogenesis (*Artigas et al., 2014*; *Greenblatt et al., 2022*). Runx2 and OSX activation has also been demonstrated to be mediated by the p38 MAPK signaling pathway in addition to the ERK1/2 signaling pathway (*Artigas et al., 2014*). *Nie et al. (2015)* showed that using a particular p38 MAPK inhibitor to disrupt the p38 MAPK signaling pathway resulted in a considerable reduction of PDLSCs' capacity for osteogenic differentiation. This implies that PDLSCs' osteogenic development is impacted by the p38 MAPK signaling pathway. *Kaneko et al. (2022)* found that the JNK inhibitor SP600125 promoted the formation of mineralized nodules and the expression of osteogenesis-related genes in hPDLSCs, inversely suggesting that the JNK signaling pathway is involved in the osteogenic differentiation of PDLSCs. Among the MAPK signaling pathways, the ERK5 signaling pathway is relatively recent, and research has indicated its involvement in PDLSC osteogenesis. *Jiang et al. (2016)* came to the conclusion that the osteogenic differentiation of hPDLSCs produced by bone morphogenetic protein (BMP) runs through the ERK5 signaling pathway.

In addition to the MAPK signaling pathway, the signaling pathways involved in cellular osteogenic differentiation include the Wnt/β-catenin signaling pathway, the BMP signaling pathway, and the Notch signaling pathway. The differences between the MAPK signaling pathway and these pathways are: the Wnt/β-catenin signaling pathway focuses more on maintaining osteoblasts differentiation and promoting bone formation in the later stages of osteogenesis, and its signaling is relatively stable and long-lasting, whereas the MAPK signaling pathway is more rapid and short-lived; the activation of the BMP signaling pathway relies primarily on specific ligand–receptor binding, whereas the MAPK signaling pathway can be activated by many different types of stimuli; the Notch signaling pathway

mainly transmits signals through direct cell-to-cell contact, whereas the MAPK signaling pathway is more responsive to signals such as extracellular soluble factors (*Fu et al., 2023*; *Ru, Pan & Zheng, 2023*).

## The regulatory role of the MAPK signaling pathways in the osteogenic differentiation of PDLSCs

The exact function that the MAPK signaling pathways play in the osteogenic differentiation of PDLSCs is still unknown, despite the fact that their involvement has been confirmed. Numerous investigations have demonstrated that the MAPK signaling pathways affect PDLSC osteogenic development in a bidirectional manner, which can be either favorably or negatively regulated. One or more subfamilies of the MAPK signaling pathways are activated when PDLSCs are subjected to different stimuli. If these pathways promote the osteogenic differentiation of PDLSCs, they are positively regulated; if they restrain PDLSCs' osteogenic differentiation, they are negatively regulated.

### MAPK signaling pathways positively regulate osteogenic differentiation of PDLSCs

*Traditional Chinese medicine ingredients promote osteogenic differentiation of PDLSCs through the MAPK signaling pathways.* Myricetin is a flavonoid found in berries, and *Kim, Park & Choung (2018)* concluded that myricetin was able to upregulate the expression of osteopontin (OPN), osteocalcin (OCN), and collagen type- I (COL1A1) through the MAPK signaling pathways, thereby promoting the osteogenic differentiation of PDLSCs. The JNK signaling route was active later during this phase, whilst the ERK and p38 signaling pathways were activated preferentially in PDLSCs. The different MAPK pathways were not activated concurrently throughout this process. Additionally, Kim found that the extent to which myricetin promotes the expression of different osteogenic proteins in PDLSCs through the MAPK signaling pathways varies. The findings demonstrated that whereas OPN and COL1A1 expression levels were only marginally raised, OCN expression levels were considerably higher.

Lycium barbarum is a common Chinese herb, and Lycium barbarum polysaccharide-glycoprotein (LBP) is its key active ingredient. *Lai et al. (2023)* suggested that LBP could promote osteogenesis in PDLSCs by activating the ERK signaling pathway. The experimental results showed that LBP can stimulate periodontal bone formation in rats and has a therapeutic effect on alveolar bone resorption caused by periodontitis. Additionally, they discovered that PDLSCs treated with LBP had higher expression levels of genes and proteins linked to the ERK pathway. The ERK signaling pathway inhibitor SCH772984 inhibited the upregulation of osteogenic differentiation-related proteins such as alkaline phosphatase (ALP) and COL1A1 induced by LBP. These findings imply that LBP promotes periodontal osteogenesis by activating the ERK signaling pathway.

Berberine is an active ingredient extracted from Coptidis rhizoma that promotes hPDLSCs in the early, middle, and late stages of osteogenesis. Through the epithelial growth factor receptor (EGFR), berberine can bind to hPDLSCs and trigger the ERK signaling cascade, upregulating the nuclear-related gene FOS and hastening the osteogenic differentiation of hPDLSCs (*Liu et al., 2018*).

Asiatic acid is derived from *Centella asiatica*. Asiatic acid has been demonstrated to have an impact on the canonical Wnt/β-catenin pathway. The ERK signaling pathway and the canonical Wnt/β-catenin pathway interact through crosstalk. Ras is a downstream signaling protein of the ERK signaling pathway, and β-catenin prevents its degradation. This indirectly activates the ERK signaling pathway, promoting osteogenic differentiation of PDLSCs and increasing BMP2 and OSX expression (*Thamnium et al., 2023*).

Another study suggests that asarylaldehyde found in the Acorus calamus promotes osteogenic differentiation of PDLSCs at non-cytotoxic concentrations (100 ng/mL) through the ERK1/2 and p38 MAPK signaling pathways (*Hwang, Park & Han, 2021*). Through the ERK1/2 signaling pathway, naringin, which is found in grapefruit peel, increases the osteogenic differentiation of PDLSCs by promoting ALP activity, Runx2 and OCN expression, and the formation of mineralized nodules (*Wei et al., 2017*). In recent years, more and more herbs have been found to promote osteogenesis in PDLSCs through the MAPK signaling pathways. Investigating the mechanisms by which these compounds promote osteogenesis may provide new ideas for adjuvant therapy for periodontitis.

*Hormones promote osteogenic differentiation of PDLSCs through the MAPK signaling pathways.* 20-Hydroxyecdysone is a polyhydroxylated steroid invertebrate hormone that belongs to the large ecdysteroid family. *Jian et al. (2013)* treated PDLSCs with 20-hydroxyecdysone and found that it increased BMP2 mRNA and protein expression. Alizarin red staining showed that PDLSCs formed more mineralized nodules after two weeks of 20-hydroxyecdysone treatment. From these results, it can be inferred that 20-hydroxyecdysone may stimulate osteogenic differentiation by inducing BMP2 expression in PDLSCs. Jian then examined the effects of the ERK1/2 inhibitor PD98059, the p38 inhibitor SB203580, and the JNK inhibitor SP600125 on 20-hydroxyecdysone-induced BMP2 mRNA and protein expression, finding that PD98059 blocked the expression of BMP2 mRNA and protein induced by 20-hydroxyecdysone, whereas SB203580 and SP600125 did not exhibit this effect. This suggests that the ERK signaling pathway is responsible for inducing BMP2-dependent osteogenic differentiation of PDLSCs by 20-hydroxyecdysone.

Furthermore, *Ge et al. (2019)* showed that the ERK signaling pathway allows oxytocin to cause PDLSCs to differentiate into osteogenic tissues. Their study found that oxytocin caused a significant increase in the phosphorylation of ERK within two minutes, reaching its highest level after five minutes. The highest levels of ALP, COL1A1, Runx2, OCN, and OPN gene expression and mineralized nodule formation were observed in PDLSCs in the presence of 50 nM oxytocin.

*Physical factors promote osteogenic differentiation of PDLSCs via the MAPK signaling pathways.* During orthodontic treatment and chronic periodontitis, excessive mechanical forces or chronic inflammation can severely damage the periodontal vascular system, leading to a hypoxic environment for PDLSCs. According to several studies, PDLSCs' osteogenic development is positively regulated by this hypoxic environment (*Zhou, Fan & Xiao, 2014*). *Wu et al. (2013)* observed that the hypoxic environment significantly activated the ERK and p38 MAPK signaling pathways in PDLSCs, stimulated the expression of

Runx2 and Sp7 transcription factors, and subsequently upregulated ALP, prostaglandin E2 (PGE2), and vascular endothelial growth factor (VEGF). The findings indicate that PDLSCs' osteogenic differentiation is encouraged by the hypoxic environment *via* the MAPK signaling pathways. This offers a mechanistic explanation for the hypoxia-induced osteogenesis and angiogenesis observed in periodontal models.

Nitric oxide (NO), *via* the JNK MAPK signaling pathway, has been demonstrated to sustain the equilibrium between osteoblasts and adipocytes in PDLSCs (*Yang et al., 2018*). Adipogenesis-induced genes such as lipoprotein lipase (LPL), peroxisome proliferator-activated receptor γ2 (PPARγ2), and CCAAT-enhancer binding protein (C/EBP) are downregulated in PDLSCs when NO is introduced, whereas ALP, Runx2, OPN, and OSX expression is upregulated. The upregulation of adipogenic factors and the downregulation of osteogenic factors caused by NO were observed when the JNK signaling pathway was inhibited. This suggests that *via* the JNK signaling pathway, NO increases osteogenic differentiation and decreases adipogenic differentiation of PDLSCs.

In recent years, light-emitting diodes (LEDs) have gradually gained attention as novel light sources for phototherapy in periodontal therapy. *Yamauchi et al. (2018)* experimentally concluded that 650 nm red high-power LED irradiation could increase the expression of Runx2 and OSX, as well as ALP activity and calcium deposition in PDLSCs. They subsequently found that LED-induced ERK1/2 activation was reduced by ERK1/2 inhibitors, which diminished the promotion of PDLSC osteogenesis by LED. According to these findings, PDLSCs' osteogenic differentiation is improved by 650 nm high-power red LED irradiation *via* the ERK1/2 signaling pathway.

Researchers also found that by triggering the p38 MAPK signaling pathway in hPDLSCs, gold nanoparticles enhanced the expression of ALP, RUNX2, OSX, and COL1A1 (*Niu et al., 2017*). According to other studies, the ERK1/2 signaling pathway may be used by magnetic stimulation to cause PDLSCs to undergo osteogenic differentiation (*Peluso et al., 2021*).

*Other factors promote osteogenic differentiation of PDLSCs through the MAPK signaling pathways.* Transforming growth factor beta (TGF-β) controls cell division and encourages mesenchymal stem cells to differentiate early. *Li et al. (2019)* showed how TGF-β3 might stimulate hPDLSCs osteogenic differentiation *via* the p38 MAPK signaling pathway. Additionally, BMP, a member of the TGF-β superfamily, has the ability to stimulate osteogenic differentiation and encourage cell aggregation in osteogenic sites. It has been demonstrated that BMP-2, BMP-6, BMP-7, and BMP-9, out of the 14 BMPs examined *in vivo*, stimulate the expression of genes linked to bone tissue and encourage the osteogenic differentiation of hPDLSCs (*Hakki et al., 2014*; *Jiang et al., 2016*). *Jiang et al. (2016)* concluded that BMP-9 promotes osteogenic differentiation of hPDLSCs more strongly than BMP-2 and BMP-7 and increases the expression of OCN, OPN, OSX, and RUNX2 by triggering the ERK5 signaling pathway. In BMP-9-induced osteogenic differentiation of PDLSCs, Reilly discovered that the JNK inhibitor SP600125 could suppress the elevation of Runx2, ALP, OPN, and OCN expression. This finding implies that the JNK signaling pathway regulates BMP-9-induced osteogenesis in PDLSCs in a favorable manner (*Wang*

*et al., 2017*). *Ye et al. (2014)* discovered that, in BMP-9-induced osteogenic differentiation of PDLSCs, the p38 signaling pathway also has a positive regulatory function, while the ERK1/2 signaling pathway has a negative regulatory one.

Progranulin (PGRN), an antagonist of tumor necrosis factor receptor (TNFR), promotes the regeneration of bone defects in periodontitis and plays a crucial role in wound healing, anti-inflammation, and osteogenesis. *Chen et al. (2020)* came to the conclusion that PGRN stimulates the ERK1/2 and JNK signaling pathways, which in turn helps PDLSCs differentiate into osteogenic tissues. Additionally, *Chen et al. (2020)* discovered that in TNFR2-silenced PDLSCs, PGRN-induced activation of the JNK signaling pathway was diminished, whereas ERK1/2 signaling pathway activation remained unaltered. It implies that TNFR2 may be required specifically for JNK signaling pathway activation. PGRN activates the JNK signaling pathway through TNFR2, while the mechanism of ERK1/2 signaling activation remains to be explored.

Dentin matrix protein 1 (DMP1) is a non-collagenous protein that is present in dentin and belongs to the SIBLING (Small Integrin-Binding Ligand, N-linked Glycoprotein) family of proteins. By triggering the ERK signaling pathway, it controls the expression of certain genes in PDLSCs. DMP1 has been shown to activate ERK protein in PDLSCs and transport it to the nucleus, which not only upregulates the expression of ALP but also decreases the expression of Twist1, a negative osteogenic regulator. This, in turn, promotes the osteogenic differentiation of PDLSCs (*Chandrasekaran et al., 2013*).

Growth arrest-specific transcript 5 (GAS5), a long non-coding RNA overexpressed in growth-arrested cells, shows significant expression differences between undifferentiated and osteogenically differentiated PDLSCs. Research on the underlying molecular mechanism of GAS5 overexpression in PDLSCs has been demonstrated to phosphorylate JNK and p38 proteins and to increase the expression of growth differentiation factor 5 (GDF5), an osteogenesis-related factor; conversely, GAS5 knockdown has the opposite effect. The findings indicate that GAS5 stimulates PDLSCs to differentiate into osteoblasts by upregulating GDF5 *via* the p38 and JNK signaling pathways (*Yang et al., 2020*).

Histone methyltransferase PRDM9 is a histone-modifying enzyme that regulates cell proliferation, differentiation, cell cycle progression, and intracellular homeostasis maintenance through intrinsic histone methyltransferase activity or interaction with other nuclear chromatin-modifying enzymes. It was found that its downstream gene, FBLN5, promotes the expression of phosphorylated p38 MAPK, ERK1/2 and JNK in hPDLSCs, thereby increasing the content of ALP and mineralized nodules in hPDLSCs and promoting their osteogenic differentiation (*Zhao et al., 2024*).

### MAPK signaling pathways negatively regulate osteogenic differentiation of PDLSCs

The development and functional role of mesenchymal stem cells are hindered by the microenvironmental circumstances. PDLSCs' capacity to promote osteogenesis can be strongly impacted by inflammatory conditions *via* processes connected to the MAPK signaling pathways. *Mao et al. (2016)* suggested that elevated levels of IL-1β prevent PDLSCs from differentiating into osteogenic tissues by stimulating the p38 MAPK and

JNK signaling pathways. The production of BMP2, OSX, OPN, and RUNX2 in PDLSCs was shown to be restored by inhibiting the p38 MAPK signaling pathway. Similarly, in PDLSCs, BMP2, OSX, OPN, and RUNX2 expression were all increased with inhibition of JNK. However, Blocking the ERK1/2 signaling pathway did not significantly alter the inhibitory impact of IL-1β on PDLSC osteogenesis. Additionally, *Dordevic et al. (2016)* discovered that IL-17 activated the ERK1/2 and JNK signaling pathways, which in turn decreased the production of ALP and OCN in PDLSCs. He successfully restored the expression of osteogenesis-related transcription factors downregulated by IL-17 using the ERK1/2 inhibitor PD98059 and the JNK inhibitor SP600125, countering IL-17's inhibitory impact on PDLSCs' osteogenic differentiation. These outcomes suggest that IL-17 inhibits osteogenic differentiation of PDLSCs through the ERK1/2 and JNK signaling pathways. In summary, it may be possible to treat periodontitis and rebuild periodontal hard tissues with new knowledge if it is determined that MAPK signaling pathways influence the inflammatory environment's suppression of PDLSCs' osteogenic potential.

Pyrophosphate is an inorganic ion consisting of two phosphate molecules, and it is considered an inhibitor of bone mineralization. *Liang et al. (2021)* found that pyrophosphate downregulated the expression of RUNX2 and OSX in PDLSCs by phosphorylating ERK1/2, JNK, and p38 MAPK proteins, with RUNX2 expression being regulated exclusively by the p38 MAPK signaling pathway. Additionally, Liang compared the duration of activation of each MAPK signaling pathway by pyrophosphate and found that the p38 MAPK pathway was activated for the longest time. These results suggest that pyrophosphate inhibits PDLSC osteogenic development by activating the MAPK signaling pathways.

The bacterial virulence factor GroEL is part of the heat shock protein family found in almost all prokaryotes and eukaryotes and is an important molecule associated with bacterial infections and autoimmune diseases. It has been shown that GroEL initiates the JNK signaling pathway and encourages the nuclear accumulation of phosphorylated JNK proteins through the activation of Toll-like receptor 2 and Toll-like receptor 4 on the cell membranes of PDLSCs, thereby decreasing the expression of ALP, preventing the development of mineralized nodules and decreasing PDLSCs' ability to osteogenesis (*Zhang et al., 2021*).

## CONCLUSIONS

In conclusion, four subfamilies of the MAPK signaling pathways: the ERK1/2 signaling pathway, the p38 MAPK signaling pathway, the JNK signaling pathway, and the ERK5 signaling pathway can participate in the osteogenic differentiation of PDLSCs. Notably, the levels of phosphorylated ERK, phosphorylated p38, and phosphorylated JNK increased significantly during the regulation of osteogenic differentiation by the MAPK signaling pathways, whereas the total amounts of ERK, p38, and JNK proteins remained unchanged. This suggests that the pathways may accomplish signaling through the activation of MAPK proteins rather than an increase in their total amounts. Numerous studies have shown that the MAPK signaling pathways can both positively regulate and negatively inhibit PDLSC

**Table 1  Factor or drug effects on promoting osteogenic differentiation of PDLSCs through MAPK pathways.**

| Positively regulate osteogenic differentiation of PDLSCs | | | |
|---|---|---|---|
| **Factor or drug** | **Target** | **Cell source** | **Ref** |
| Myricetin | ERK1/2 JNK p38 MAPK | upregulate OPN, OCN, COL1A1 | *Kim, Park & Choung (2018)* |
| Lycium barbarum polysaccharide-glycoprotein | ERK1/2 | upregulate ALP, COL1A1 | *Lai et al. (2023)* |
| Berberine | ERK1/2 | upregulate FOS | *Liu et al. (2018)* |
| Asiatic acid | ERK1/2 | upregulate BMP2, OSX | *Thamnium et al. (2023)* |
| Asarylaldehyde | ERK1/2 p38 MAPK | upregulate ALP, DMP1, OPN, COL1A1, Runx2 | *Hwang, Park & Han (2021)* |
| Naringin | ERK1/2 | upregulate ALP, OCN, Runx2 | *Wei et al. (2017)* |
| 20-hydroxyecdysone | ERK1/2 | upregulate BMP2 | *Jian et al. (2013)* |
| Oxytocin | ERK1/2 | upregulate ALP, COL1A1, OCN, OPN, Runx2 | *Ge et al. (2019)* |
| Hypoxia | ERK1/2 p38 MAPK | upregulate ALP, Sp7, Runx2, PGE2, VEGF | *Wu et al. (2013)* |
| Nitric oxide | JNK | upregulate ALP, Runx2, OPN, OSX downregulateLPL, PPAR$\gamma^2$, C/EBP | *Yang et al. (2018)* |
| Light-emitting diode | ERK1/2 | upregulate ALP, OSX, OCN, Runx2 | *Yamauchi et al. (2018)* |
| Gold nanoparticles | p38 MAPK | upregulate ALP, COL1A1, OSX, RUNX2 | *Niu et al. (2017)* |
| Magnetic Stimulation | ERK1/2 | upregulate ALP | *Peluso et al. (2021)* |
| TGF-$\beta$3 | p38 MAPK | upregulate ALP | *Li et al. (2019)* |
| BMP-2/6/7/9 | JNK p38 MAPK ERK5 | upregulate ALP, OCN, OPN, OSX, RUNX2 | *Hakki et al. (2014)*, *Jiang et al. (2016)*, *Wang et al. (2017)*, *Ye et al. (2014)* |
| Progranulin | ERK1/2 JNK | upregulate ALP, RUNX2 | *Chen et al. (2020)* |
| Dentin matrix protein 1 | ERK1/2 | upregulate ALP down-regulateTwist1 | *Chandrasekaran et al. (2013)* |
| Growth arrest specific transcript 5 | JNK p38 MAPK | upregulate growth differentiation factor 5 | *Yang et al. (2020)* |
| Histone methyltransferase PRDM9 | ERK1/2 JNK p38 MAPK | upregulate ALP | *Zhao et al. (2024)* |

osteogenesis. Additionally, the timing of the effects of the various enzymes involved in signaling regulation may differ, making the role of these pathways in PDLSC osteogenesis controversial.

However, the regulation of the MAPK signaling pathway to regulate the osteogenic differentiation of PDLSCs still has many limitations. Most experiments on the MAPK signaling pathways regulating the osteogenic differentiation of PDLSCs have been limited to basic cellular studies. These studies verify that some factors regulate the osteogenic differentiation of PDLSCs *via* the MAPK signaling pathways (Tables 1 and 2, Fig. 3), but the mechanisms by which these pathways regulate PDLSC osteogenic differentiation are not yet fully understood. MAPK signaling pathway interactions are complex; the

**Table 2 Factor or drug effects on reducing osteogenic differentiation of PDLSCs through MAPK pathways.**

| Negatively regulate osteogenic differentiation of PDLSCs | | | |
|---|---|---|---|
| Factor or drug | Target | Cell source | Ref |
| IL-1β | JNK p38 MAPK | **downregulate** BMP2, OSX, RUNX2, OPN | *Mao et al. (2016)* |
| IL-17 | ERK1/2 JNK | **downregulate** ALP, OCN | *Dordevic et al. (2016)* |
| Pyrophosphate | ERK1/2 JNK 38 MAPK | **downregulate** RUNX, OSX | *Liang et al. (2021)* |
| GroEL | JNK | **downregulate** ALP | *Zhang et al. (2021)* |

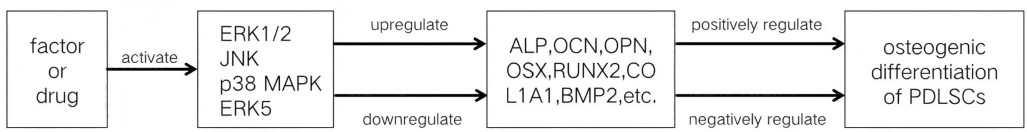

**Figure 3 Factor or drug effects on osteogenic differentiation of PDLSCs through MAPK pathways.**

activation of one signaling pathway may affect the activity of other pathways, and it is difficult to accurately regulate a certain pathway to achieve the desired effect of osteogenic differentiation, which increases the difficulty and uncertainty of regulation. If future studies can clarify the targets of each step in MAPK signaling pathway activation by different factors in regulating PDLSC osteogenic differentiation, it will provide an important theoretical basis for periodontal regenerative tissue engineering based on PDLSCs.

### Funding
This work was approved by grants from the Clinical Medicine Talents Training Program of the Hebei Provincial Government (361029). The funders had no role in study design, data collection and analysis, decision to publish, or preparation of the manuscript.

### Grant Disclosures
The following grant information was disclosed by the authors:
Clinical Medicine Talents Training Program: 361029.

### Competing Interests
The authors declare there are no competing interests.

### Author Contributions
- Xuanning Liu conceived and designed the experiments, performed the experiments, prepared figures and/or tables, and approved the final draft.

- Wanqing Zhao conceived and designed the experiments, performed the experiments, prepared figures and/or tables, and approved the final draft.
- Yanhui Peng performed the experiments, prepared figures and/or tables, and approved the final draft.
- Na Liu analyzed the data, authored or reviewed drafts of the article, and approved the final draft.
- Qing Liu analyzed the data, authored or reviewed drafts of the article, and approved the final draft.

## Data Availability

This is a literature review.

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
