# Peer review of "The relationship between MAPK signaling pathways and osteogenic differentiation of periodontal ligament stem cells: a literature review"

_PeerJ, doi:10.7717/peerj.19193_

## Round 0.1 · original submission · Major Revisions

This is an interesting and relevant article. Please address the reviewer's comments and resubmit the revised manuscript.

·

Basic reporting

See below

Experimental design

See below

Validity of the findings

See below

Additional comments

This paper reviews the relationship between the MAPK 73 signaling pathways and osteogenic differentiation of PDLSCs, summarizes the currently known 74 relevant factors that affect the osteogenic differentiation of PDLSCs by activating the MAPK 75 signaling pathways, and provides research ideas for the better application of PDLSCs in 76 periodontal tissue regeneration. I think you will find this review very interesting. We suggest adding more information for 1.1, 1.2, 1.3, and 1.4. in Biological effects of the MAPK signaling pathways section.

Reviewer 2 ·

Basic reporting

The manuscript is well-written and follows a clear structure, but it would benefit from updated references to include recent studies. Figures and tables are informative but could use better labeling and detailed legends for clarity. Adding a summary table or schematic diagram would help present complex data more effectively. Overall, the content is relevant and provides valuable insights into MAPK signaling and osteogenesis.

Experimental design

The study design is appropriate for a review article, with a clear focus on MAPK signaling pathways and their role in osteogenesis. However, the methodology for selecting and analyzing studies is not sufficiently detailed. Providing explicit inclusion and exclusion criteria for the reviewed literature would enhance transparency and reproducibility. Additionally, integrating a comparative analysis of MAPK signaling with other related pathways would strengthen the review's impact.

Validity of the findings

The findings in the manuscript are valid and supported by the cited literature. However, the review would benefit from including more recent studies to ensure the conclusions are up-to-date. The analysis of MAPK signaling pathways is thorough, but some key gaps, such as potential adverse effects or limitations of therapeutic applications, need to be addressed to provide a balanced perspective. Overall, the findings are credible but could be strengthened with additional critical analysis and integration of newer evidence.

Additional comments

No

---

## Round 0.2 · accepted · Accept

Thank you for addressing the comments from the reviewers.

·

Basic reporting

no comment

Experimental design

no comment

Validity of the findings

no comment

Additional comments

no comment

Reviewer 2 ·

Basic reporting

The revisions have successfully addressed the reviewers' concerns.
I recommend that PeerJ accept the manuscript for publication.

Experimental design

The revisions have successfully addressed the reviewers' concerns.
I recommend that PeerJ accept the manuscript for publication.

Validity of the findings

The revisions have successfully addressed the reviewers' concerns.
I recommend that PeerJ accept the manuscript for publication.

Additional comments

The revisions have successfully addressed the reviewers' concerns.
I recommend that PeerJ accept the manuscript for publication.